# Interaction between Mas1 and AT1R$_A$ contributes to enhancement of skeletal muscle angiogenesis by angiotensin-(1-7) in Dahl salt-sensitive rats

Eric C. Exner[ORCID][1], Aron M. Geurts[1,2], Brian R. Hoffmann[1,3,4], Marc Casati[4], Timothy Stodola[1], Nikita R. Dsouza[2], Michael Zimmermann[2], Julian H. Lombard[1], Andrew S. Greene[5]*

1 Department of Physiology, Medical College of Wisconsin, Milwaukee, Wisconsin, United States of America, 2 Genomic Sciences and Precision Medicine Center, Medical College of Wisconsin, Milwaukee, Wisconsin, United States of America, 3 Department of Bioengineering, Medical College of Wisconsin and Marquette University, Milwaukee, Wisconsin, United States of America, 4 Cardiovascular Center, Medical College of Wisconsin, Milwaukee, Wisconsin, United States of America, 5 The Jackson Laboratory, Bar Harbor, Maine, United States of America

* Andy.Greene@jax.org

**Data Availability Statement:** All relevant data are within the manuscript and its Supporting Information files.

## Abstract

The heptapeptide angiotensin-(1–7) (Ang-(1–7)) is protective in the cardiovascular system through its induction of vasodilator production and angiogenesis. Despite acting antagonistically to the effects of elevated, pathophysiological levels of angiotensin II (AngII), recent evidence has identified convergent and beneficial effects of low levels of both Ang-(1–7) and AngII. Previous work identified the AngII receptor type I (AT1R) as a component of the protein complex formed when Ang-(1–7) binds its receptor, Mas1. Importantly, pharmacological blockade of AT1R did not alter the effects of Ang-(1–7). Here, we use a novel mutation of AT1R$_A$ in the Dahl salt-sensitive (SS) rat to test the hypothesis that interaction between Mas1 and AT1R contributes to proangiogenic Ang-(1–7) signaling. In a model of hind limb angiogenesis induced by electrical stimulation, we find that the restoration of skeletal muscle angiogenesis in SS rats by Ang-(1–7) infusion is impaired in AT1R$_A$ knockout rats. Enhancement of endothelial cell (EC) tube formation capacity by Ang-(1–7) is similarly blunted in AT1R$_A$ mutant ECs. Transcriptional changes elicited by Ang-(1–7) in SS rat ECs are altered in AT1R$_A$ mutant ECs, and tandem mass spectrometry-based proteomics demonstrate that the protein complex formed upon binding of Ang-(1–7) to Mas1 is altered in AT1R$_A$ mutant ECs. Together, these data support the hypothesis that interaction between AT1R and Mas1 contributes to proangiogenic Ang-(1–7) signaling.

## Introduction

Microvascular dysfunction is an important risk factor for highly prevalent cardiovascular diseases including hypertension and diabetes. The renin-angiotensin system (RAS) plays a key

**Funding:** Sources of Funding: This study was supported by National Institutes of Health grants F30-HL131153, P01-HL116264, R01-HL125409, R01-HL128242, R21-OD024781, K01-DK105043 and the Kern Family Foundation. Eric C. Exner is a member of the Medical Scientist Training Program at the Medical College of Wisconsin, which is partially supported by a training grant, NIGMS T32-GM080202. The funders had no role in study design, data collection and analysis, decision to publish, or preparation of the manuscript. Kern Family Foundation: https://www.kffdn.org/ National Institutes of Health: https://www.nih.gov/.

**Competing interests:** The authors have declared that no competing interests exist.

role in maintaining microvascular function. Systemic manipulation of the RAS, especially by interfering with the signaling of angiotensin II (AngII) at the angiotensin II receptor type 1 (AT1R), is now a mainstay of treatment in cardiovascular disease. The success of drug treatments targeting the RAS has suggested that signaling via AT1R is predominantly associated with negative health outcomes. However, recent studies reaffirm that AngII/AT1R signaling plays a key role in vascular homeostasis. For instance, renin suppression in animal models impairs angiogenesis [1] and vascular reactivity to vasodilator stimuli[2], and restoration of physiological AngII levels in models of impaired renin regulation restores vascular reactivity and endothelial function [3,4].

While AngII is the canonical effector peptide of the RAS, other RAS peptides have become increasingly interesting as avenues for therapeutic intervention. Foremost among these is angiotensin-(1–7) (Ang-(1–7)), a RAS effector peptide with effects mediated by the receptor Mas1 [5].

Ang-(1–7) has been shown to exhibit several protective cardiovascular effects that act antagonistically to pathophysiological elevations in AngII levels. These effects are wide-ranging, including rescue of endothelial dysfunction and promotion of antifibrotic/antihypertrophic phenotypes [6,7]. Conversely, under conditions of RAS suppression (i.e. low-renin hypertension or salt-induced renin-suppression) restoration of physiological AngII levels and low-dose Ang-(1–7) have convergent, beneficial effects such as improvement of endothelial function and rescue of impaired vasodilator responses[3].

The Ang-(1–7)/Mas axis of the RAS is relatively understudied, especially compared to the AngII/AT1R axis, but recent studies have identified several key mediators of Ang-(1–7)/Mas signaling. Several of these pathways are also important in AngII/AT1R signaling. For instance, Ang-(1–7) contributes to endothelial homeostasis via ERK1/2, eNOS, PI3-kinase and Akt, all of which are also associated with AngII signaling via AT1R [4,8–10]. A few relatively detailed proteomic analyses of Ang-(1–7)/Mas signaling have been performed in our laboratory and by others, and the results of those studies further support a convergence in AngII/AT1R and Ang-(1–7)/Mas signaling [3,11].

In a previous effort to define proteins important to the transduction of Ang-(1–7)/Mas signaling, we identified AT1R as a component of the protein complex formed when Ang-(1–7) binds to its receptor, Mas1[3]. Heterodimerization of these receptors has been suggested previously. In fact, co-localization of AT1R and the receptor Mas1 has been demonstrated by bioluminescence energy transfer (BRET)[12], although dependence of Ang-(1–7)/Mas signaling on the AT1R has yet to be described[12,13]. Here, we test the hypothesis that AT1R contributes to the proangiogenic effects of Mas1 mediated Ang-(1–7) signaling. It is important to note that pharmacological blockade of AT1R via losartan does not alter the effects of Ang-(1–7) treatment[3]. Thus, the contribution of AT1R to Ang-(1–7)/Mas1 signaling in this model is independent of AT1R ligand binding.

We have established an animal model of skeletal muscle angiogenesis in which one hind limb is electrically stimulated to produce rhythmic muscular contractions over the course of seven days, resulting in increased microvessel density in angiogenically competent rats[14]. In this model, we have observed that modulation of the renin-angiotensin system affects angiogenesis. In particular, suppression of the renin-angiotensin system genetically[1,15], pharmacologically[3,15,16], or via environmental factors[1] impairs angiogenesis. We recently observed that low doses of angiotensin II and angiotensin-(1–7) have convergent, proangiogenic effects in this model of impaired angiogenesis [3].

In this study, we investigate the restoration of normal microvascular function in the Dahl salt-sensitive rat, a genetic model of low renin-angiotensin system activity and impaired angiogenesis, to test the hypothesis that proangiogenic Ang-(1–7) signaling is dependent on the

presence of AT1R. Using wild type Dahl salt-sensitive (SS-AT1WT) rats and Dahl salt-sensitive rats with a novel mutation in AT1R resulting in an early truncation and loss of function (SS-AT1KO), we assess the consistency of Ang-(1–7) induced effects at the physiological level using a hind-limb model of angiogenesis, at the cellular level via endothelial cell tube formation, and at the molecular level via qPCR and tandem mass spectrometry-based proteomics.

# Methods

## Summary

Angiogenesis was assessed in catheterized, infused rats (3.0ng/kg/min Ang-(1–7) in 0.9% saline) implanted with an electrical stimulator to induce unilateral, rhythmic contractions of muscles in one hind limb[3]. The difference in vessel density between stimulated and unstimulated limbs was used as a measure of angiogenesis. Angiogenic capability of endothelial cells was assessed using an *in vitro* tube formation assay[17]. Total length of tubes was measured using Pipeline software[18]. Gene expression was assessed by quantitative polymerase chain reaction (qPCR). Mass spectrometry based proteomics was performed to capture interacting proteins as previously described [3]. Briefly, endothelial cells were frozen in liquid nitrogen and ground into a powder. This was followed by immunoprecipitation using Mas1 antibody coupled magnetic beads, and captured proteins were subjected to liquid chromatography and tandem mass spectrometry[3].

While humans express one AT1R protein, rats and mice express two versions of the protein, angiotensin II receptor type 1a (AT1R $_A$) and type 1b (AT1R $_B$). In the rat, AT1R $_B$ is expressed in the adrenal, but not the microvasculature [19]; thus we used a novel AT1R $_A$ knockout on the background of the Dahl salt-sensitive rat along with its wild type control for these studies. Functional deletion of the AT1R response in these animals was confirmed *in vivo* via an acute blood pressure response to angiotensin II (0.32ug/kg i.v.).

## Animals and infusion of angiotensin peptides

All animal protocols were approved by the Medical College of Wisconsin (MCW) Institutional Animal Care and Use Committee. Animals were housed and cared for at the MCW Animal Resource Center. All rats were maintained on normal salt diet (0.4% sodium chloride—AIN-76A, Dyets Inc. #113755) for the duration of this study with free access to water, as previously described [2]. Wild type, male Dahl salt-sensitive rats (SS-AT$_1$WT) and male Dahl salt-sensitive rats with a novel mutation in AT1R resulting in an early truncation and loss of function (SS-AT$_1$KO) underwent 7 days of hind-limb electrical stimulation and received different treatments during the entire stimulation protocol. Rats were randomly assigned to the following groups: vehicle infusion and Ang-(1–7) (2.6 ng · kg$^{-1}$ · min$^{-1}$ i.v.). All rats completed experimental protocols at 9–12 weeks of age. Ang-(1–7) dosage was matched to previous studies in this model[3]. That dose was chosen to be equimolar to a subpressor dose of AngII used in previous studies. Using reported *in vivo* half-lives of these peptides in rat models[20] and a simple one-compartment model for the circulatory system, estimated plasma steady-state Ang-(1–7) concentration was similar to baseline Ang-(1–7) concentrations reported in age-matched, untreated Sprague-Dawley rats.

## Zinc-Finger Nuclease (ZFN) mutant generation, and genotyping

For the AT1R $_A$ receptor knockout rats, ZFN constructs targeting the sequence CTTTGCCCCTGTGGGCAGTCTATACCGCTATGGAGTACCGCT of exon 3 of the *Agtr1$_A$* gene were produced by Sigma-Aldrich (St. Louis, MO), where the underlined sequences indicate

individual ZFN monomer binding on opposite strands. Messenger RNA encoding the *Agtr1a* ZFN sequences was injected at a concentration of 10 ng/μl into the 1-cell pronucleus of SS/ JrHsdMcwi (SS) rat embryos and implanted into pseudopregnant females[21]. DNA was extracted from founder generation pups at 10 days of age and used for PCR genotyping. Founder mutants were identified by CEL-I assay and confirmed by Sanger sequencing [22] using the following primers: forward, 5′-CCTCTACAGCATCATCTTTGTGG-3′; and reverse, 5′-CACACTGGCGTAGAGGTTGA-3'. This process produced a 2-bp frame shift deletion of bases (see below) in the ZFN target sequence, corresponding to nucleotides 521–522 (TC) of reference sequence NM_030985, and resulting in 12 nonsense amino acids before the introduction of a premature stop codon. The mutant founder animal was bred to the parental strain to establish germline transmission and a colony of SS-*Agtr1a*[em1Mcwi] mutant rats (RGD ID: 5685369) was established.

## Verification of mutation by restriction digest

Verification of the mutation in the AT1R $_A$ receptor protein itself was not possible by Western blot analysis because commercially available antibodies are unable to specifically distinguish between AT1R $_A$ and AT1R $_B$ receptor subtypes[23]. Therefore, RT-PCR of *Agtr1a* cDNA followed by a restriction digest was performed. Based on Sanger sequencing results, the 2-bp deletion creates a restriction site for the enzyme *AccI* (G***TC***TATAC mutated to GTATAC) in the knockout animals, which is not present in the SS parental strain.

SS rats and *Agtr1a* homozygous knockouts were maintained on a low salt (LS; 0.4% NaCl) diet post-weaning and were sacrificed using Beuthanasia diluted in saline (final dose 195 mg/ kg pentobarbital sodium and 25 mg/kg phenytoin sodium). Tissue was obtained from the kidneys of each animal and stored in RNA Later (Life Technologies, Grand Island, NY). Total RNA was extracted from a 25 mg tissue sample (IBI Scientific, Peosta, IA), and a volume of RNA equal to 800 ng was reverse transcribed to cDNA using an AffinityScript cDNA Synthesis Kit from Agilent Technologies (Santa Clara, CA) according to manufacturer's instructions.

RT-PCR was performed with a Stratagene Mx3000P qPCR machine (Agilent Technologies, Santa Clara, CA) using the cDNA product as a template run. Each 25 μl reaction contained 1.0 μl cDNA, 0.5 μl forward primer (10 μM), 0.5 μL reverse primer (10 μM), 12.5 μL RT$^2$ SYBR Green qPCR Mastermix, and 10.5 μL RNase-free water. The RT$^2$ SYBR Green qPCR Mastermix was obtained from Qiagen (Valencia, CA). AT1R$_A$ primers were: forward, 5′-GGAAA-CAGCTTGGTGGTGAT-3′; and reverse, 5′-ACATAGGTGATTGCCGAAGG-3′. The thermal profile was 95˚C denaturation for 10 minutes, followed by 40 cycles of 95˚C for 15 seconds and 60˚C for 60 seconds.

The *AccI* restriction digest to differentiate between *Agtr1a* wild type and knockout animals was performed as follows: each 50 μL reaction was composed of 1 μL *AccI* enzyme (New England Biolabs, Ipswich, MA), 5 μL 10X NEBuffer 4, 15 μL RT-PCR product, and 29 μL ddH$_2$O. The reactions were digested at 37˚C for 90 minutes. In these studies, the wild type AT1R$_A$ and AT1R$_B$ receptors should show a single band at 171 bp in the presence of *AccI*, while the AT1R$_A$ knockout samples would show the 171 bp band for the AT1R$_B$ receptor and two smaller bands (125 bp and 44 bp) for the AT1R$_A$ receptor because of the *AccI* restriction site introduced by the mutation. The digest product or undigested control samples lacking the *AccI* enzyme were run on a Tris-HCl Criterion Precast 15% polyacrylamide gel (Bio-Rad, Hercules, CA) at 240V for 45 minutes. The gel was post-stained with ethidium bromide before being imaged with UV light.

## Anesthetized blood pressure recording

A Tygon catheter was implanted in the carotid artery of anesthetized (0.8L/min isoflurane) SS-AT$_1$WT and SS-AT$_1$KO rats. Blood pressure was measured in the carotid artery using a SPR-838 Millar Mikro Tip catheter (Millar Instruments). Blood pressure analysis was completed with WINDAQ software (DATAQ Instruments). A 100uL bolus of saline and/or a 100uL bolus of saline containing 0.32μg/kg AngII was administered via tail vein injection.

## rt-PCR analysis of EC receptor expression

Endothelial cells were lysed with TRIzol (Invitrogen, 1 mL per 100mm plate). RNA was isolated as previously described[24]. Purified RNA from sample was quantified with Nanodrop spectrophotometer (Thermo Scientific) and run on a QuantStudio 6 Flex real-time PCR machine (Applied Biosystems). Samples were run with the Taqman Fast Virus 1-step kit (Applied Biosystems) per manufacturer's instructions. The following gene expression assays were used: Mas1 (Invitrogen 444889), AGTR1A (IDT Rn.PT.56a.18540744), and AGTR1B (IDT Rn.PT.56a.37062306)

## Immunoblotting

SS-AT$_1$WT and SS-AT$_1$KO rat cardiac microvascular endothelial cells (RMVECs) (Cell Biologics) grown according to company protocol were brought to 70–90% confluency. Media was aspirated and cells were scraped in MPER buffer (Pierce 78501) containing Protease Inhibitor (Roche 11697498001. Cells were lysed with a 21-gauge needle and assayed for protein concentration with MicroBCA kit (Pierce 23235). Thirty-five micrograms of protein from each sample was loaded on a 10% TGX PAGE gel (Biorad) and transferred to nitrocellulose. Blot was blocked overnight in 5% NFDM (Biorad 1706404) and 1% BSA (Sigma A7906). Blot was incubated with Mas1 primary antibody (Santa Cruz sc-135063) at 1:1000 dilution overnight at 4 degrees C. Blot was rinsed and incubated with goat anti-rabbit HRP-conjugated secondary antibody (Biorad 1706515) at 1:5000 dilution for 1 hour. Blot was visualized with SuperSignal West Pico Chemiluminescence Substrate (Pierce 34080). Membrane was imaged on ImageQuant LAS 500 (GE Healthcare Life Sciences) and images were analyzed using ImageJ software (https://imagej.nih.gov/ij/).

## Electrical stimulator surgical procedures

Electrical stimulator and jugular catheters were implanted as previously described[25]. Anesthetized rats received subcutaneous incisions over the thoracolumbar region and medial aspect of the right leg, and a miniature battery powered stimulator was implanted as previously described[14]. Incisions were also made in the ventral and dorsal thoracic regions. A Tygon catheter was implanted in the jugular vein, tunneled subcutaneously, and exteriorized at the back of the neck. The catheter was passed through metal spring to a swivel allowing the animal full range of motion. After 24 h of recovery, continuous infusion of Ang-(1–7) or saline vehicle was started at a rate of 0.12 ml/h as noted above. The stimulator was activated by magnetic reed switch and electrodes located near the common peroneal nerve in the lower leg produced square-wave impulses of 0.3 ms duration, 10-Hz frequency and 3-V potential, causing intermittent contractions of the tibialis anterior (TA) muscles for eight consecutive hours, daily for the remainder of the study. The contralateral leg was used as a control and all animals were euthanized after seven days of stimulation, followed by collection of the TA for morphological analysis.

## Tissue harvest and morphological analysis of vessel density

Animals were euthanized by an overdose of Beuthanasia solution, and the stimulated and contralateral unstimulated TA muscles were excised and weighed. Muscles were fixed overnight in a 0.25% formalin solution, microsectioned, and immersed in a solution of 30μg/mL rhodamine labeled *Griffonia simplicifolia* I (GS-I) lectin (Vector Labs) for 2 hours. The sections were rinsed, mounted on microscope slides, and visualized with a fluorescent microscope system (Nikon E-80i microscope with Q-Imaging QIClick camera, 200x)[14,25]. Images were taken from at least twenty representative fields from each muscle and analyzed using Metamorph software (Molecular Devices) for percent change in microvessel density[14,25]. Vessel counts from all fields were averaged to a single vessel density defined as the mean number of vessel-grid intersections per microscope field (0.155 mm$^2$) for each muscle. Within experimental groups mean vessel densities of stimulated muscles were compared to contralateral unstimulated muscles, presented as mean ± SE, and evaluated using a paired *t*-test.

## Angiogenesis tube formation assay

Tube formation assay was performed as in previous studies[3], with the exception of the slide format used. SS-AT$_1$WT and SS-AT$_1$KO rat cardiac microvascular endothelial cells (RMVECs) (Cell Biologics) grown according to company protocol were brought to 70–90% confluency, washed twice with DPBS, and lifted using Enzymatic Free Cell Dissociation Buffer (Millipore) with gentle agitation for 30 minutes at 37˚C. RMVECs were then centrifuged at 300 x g for 5 minutes, washed twice with DPBS, resuspended in 1 mL MCDB131 basal media plus 2% FBS, and counted using the cell Countess system (Invitrogen). RMVECs were diluted for the addition of 1,250 cells in 50uL of media per well of u-Angiogenesis slides (iBidi) coated in 11 uL of Geltrex (Thermo Fisher). Serum-starved and growth factor depleted conditions were utilized as a tool to stunt normal RMVEC tube formation stimulation to better decipher changes that would be observed through the addition of 100 nM Ang-(1–7). Treatment conditions included RMVECs plus vehicle and RMVECs plus 100 nM Ang-(1–7). At 24 and 48 hours 10X magnification images were taken using a TS100 Inverted Microscope (Nikon Corporation) for analysis of the mean tube length per field (μm) using open-access PipeLine tube formation analysis software[18]. The results were averaged across biological and technical replicates followed by 1-way ANOVA.

## Isolation of Mas1 receptor signal protein complex

Immunoprecipitation was performed as in the previous study[3]. Previously, Mas1 immunoprecipitation conditions without dithiobis(succinimidyl propionate) (DSP) crosslinker were determined to be optimal due to DSP epitope inhibition[3]; therefore cryolysis was used in place of crosslinking during the immuoprecipitations (IP) to stabilize the protein complex. RMVECs were divided into 100 nM Ang-(1–7) treated (30 minutes at 37˚C) or non-treated groups, washed 3 times with ice cold DPBS (non-treated) or DPBS plus 100 nM Ang-(1–7) (treated), cells were scraped, and supernatants were transferred to 50 mL conical tubes. Following centrifugation at 300 x g for 10 minutes at 4˚C, supernatant was aspirated, washed with 10 mL ice cold DPBS plus or minus Ang-(1–7) as before, and the process was repeated twice. RMVECs were resuspended in 20 mM Hepes/1.2% PVP buffer with protease inhibitors and kept on ice until the next step. Cryolysis of the RMVECs was then performed in liquid nitrogen according to the Life Technologies cryolysis protocol in the Dynabeads Co-IP Kit (cat. #143.21D). Frozen cell pellets were placed in a liquid nitrogen cooled 2 mL microcentrifuge tube with a sterile metal bead, frozen tubes were secured in an oscillating homogenizer, and samples were oscillated at 30 hertz for 1 minute three times or until a powder is formed; tubes

were re-submerged in liquid nitrogen between each oscillation. Frozen cell pellet powders were then resuspended in solution with anti-Mas1antibody (Santa Cruz; cat. #sc-135063) coupled Dynabeads, incubated 30 minutes at 4˚C in a thermomixer at 500 rpm, and immuno-precipitated according to the Invitrogen Dynabeads Co-Immunoprecipitation Kit (cat. #143.21D) and M-270 Epoxy Dynabeads Antibody Coupling Kit (cat. #143.11D) manufacturer protocols.

## Liquid chromatography and mass spectrometry (ms) analysis

Liquid chromatography and mass spectrometry analysis was performed as in the previous study[3]. Isolated protein samples were dried using a vacuum centrifuge, resuspended in 100 μL 25 mM ammonium bicarbonate, and prepared for LC-MS/MS as described previously [26]. Tryptic peptide mixtures (1.9 μl) were separated using a NanoAccuity UPLC system (Waters, Milford, MA) coupled with an in-house packed 5Å C18 resin (Phenomenex, Torrance, CA) column (15 cm long 50μm inner diameter). A 120 minute gradient from 98% HPLC water/2% ACN/0.1% formic acid to 98% ACN/2% HPLC water/0.1% formic acid was used and peptides were analyzed using an LTQ-Orbitrap Velos MS(Thermo Scientific, Waltham, MA). All Orbitrap Velos MS/MS settings utilized were as indicated in our previous studies[26,27]. Raw mass spectra were searched against a Uniprot Rodent Database in both SEQUEST and MASCOT search algorithms, from which the best match for each scan was kept after combining searches for individual runs. Variable modification of +57-Da for alkylation of cysteine and +16-Da for oxidation of methionine were included in search parameters. Utilizing in-house Visualize proteomic analysis software[28], protein matches were filtered to remove redundancies, to remove common contaminants, selected for a P$\geq$0.95 (FDR<5%), and a comparison of groups was then run on the Ang-(1–7) treated versus non-treated immuno-precipitation protein data. Further filters were applied to the comparison, including significant increase in the Ang-(1–7) treated sample (p$\leq$0.05), presence in $\geq$ 4 (of 6) runs within a single biological group, and a fold change of at least 2 in the treated group compared to the untreated group. This dataset was used for subsequent pathway mapping using a combination of UniprotKB, StringDB, Ingenuity Pathway Analysis, and Protein Center platforms.

## Statistical analysis of MS/MS comparisons

All MS/MS statistical analyses were performed utilizing open source Visualize software with built-in statistical analysis for large proteomic dataset comparison[28]. These analyses utilized the G test, which is a log-likelihood ratio test, whose distribution can be approximated by a chi-squared distribution with a single degree of freedom[29]. For null hypotheses we assume that the expected proportion of scans for a given protein is directly related to the ratio of the total scans in each group. Each observed scan count for each protein is multiplied by this ratio, or its inverse depending on which group, to give us our expected frequency. Observed scans from Group 0 (S0) are multiplied by the ratio of total scans in Group 1 (T1) over Total Scans in Group 0 (T0) (E0 = S0*T1/T0), likewise observed scans from Group 1 (S1) are multiplied by the ratio of T0 over T1 (E1 = S1*(T0/T1)). The specific G value calculation is 2 * (S0 * ln(S0/E0) + S1 * ln(S1/E1)), and thus if our observed frequencies perfectly fit our expected frequencies, we would get a G-value of 0, and a larger G-value the more our observed frequency depart from our expected frequency. The distribution of G can be approximated by a chi-squared distribution with a degree of freedom of one to determine significance (P < 0.05), as described previously[30].

### Real-Time PCR analysis of RMVECs

The RT$^2$ Profiler$^{TM}$ PCR Array PARN-024Z (QIAGEN) designed for profiling the expression of 84 common angiogenesis related genes was used to examine expression changes induced in RMVECs by Ang-(1–7). Comparisons were made between Ang-(1–7) stimulated and non-stimulated RMVECs. One 100 mm plate of RMVECs was incubated plus/minus 100 nM Ang-(1–7) in DPBS for 2 hours at 37˚C, scraped, and RNA was isolated using the RNeasy Mini-Kit (QIAGEN, cat. #74104) according to manufacturer protocol. The RNase-Free DNase Set (QIAGEN, cat. # 79254) was used for elimination of DNA contamination. Isolated RNA concentration was measured using absorbance on the NanoDrop System. ~400–700 ng of RNA was then converted to cDNA using the RT$^2$ First Strand Kit (QIAGEN, cat. # 330401) and diluted according the manufacturer protocol for the RT$^2$ Profiler$^{TM}$ PCR Array PARN-024Z (QIAGEN). RT$^2$ SYBR Green ROX$^{TM}$ (QIAGEN, cat. # 330522) was used for the array. Samples were run in the QuantStudio 6 Flex (Applied Biosystems), thresholded and normalized according to manufacturer protocol, and statistically analyzed using the QIAGEN online RT$^2$ Profiler PCR Array Data Analysis Software version 3.5. RT-PCR analyses were then compared with the proteomic pathway analysis data to formulate the influence of Ang-(1–7) on signaling in RMVECs. Gene and protein lists were then analyzed using a combination of Ingenuity Pathways Analysis software, UniprotKB, and Protein Center software to develop pathways incorporating the data. Pathway figures were produced using Servier Medical Art (www. servier.com).

### Statistics and data analysis

Vessel density is presented as mean +/- standard error and was analyzed in SigmaPlot via paired Student t-test. Tube formation data are presented as mean +/- standard deviation and were analyzed in SigmaPlot via 1-way ANOVA with Dunnett's Method for multiple comparisons. qPCR data is presented as mean +/- standard deviation and analyzed in SigmaPlot via Student t-test. Gene expression array data was assessed via Deming regression and Student t-tests with samples paired within the same plate, performed in SigmaPlot; p-values were then assessed via a modified Hochberg's step-up procedure[31] with false discovery rate set at 0.05. Tandem mass spectrometry spectral data was analyzed using MASCOT, SEQUEST, and VIZUALIZE, with differences identified via G-test as previously described[3].

## Results

### Molecular verification of selective mutation of the AT1R$_A$ receptor

Fig 1 shows a polyacrylamide gel performed to verify successful mutation of the AT1R$_A$ receptor utilizing the *AccI* restriction digest. Total RNA was extracted from kidney samples taken from SS rats and homozygous AT1R$_A$ receptor mutant rats and reverse transcribed to cDNA. The cDNA was amplified with PCR using primers that did not differentiate the AT1R receptor subtypes, and then subjected to *AccI* restriction digest. Expected full-length PCR products of 171 bp for the AT1R$_A$ and AT1R$_B$ receptors were present in the SS samples. However, AT1R$_A$ samples subjected to *AccI* restriction digest showed 3 separate bands: a wild type AT1R$_B$ fragment at 171 bp and bands at 125 and 44 bp representing the expected sizes of the digested AT1R$_A$ mutant allele. In undigested cDNA control samples without *AccI*, the smaller bands present in AccI restriction digests of cDNA from the AT1R$_A$ knockout animals were absent in samples that were not exposed to the restriction enzyme.

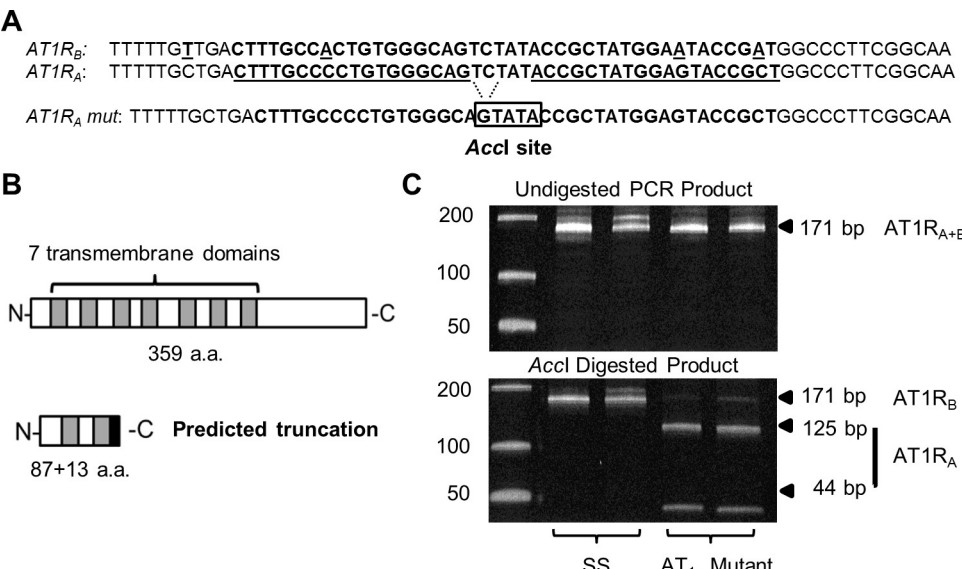

**Fig 1. Validation of the AT1R$_A$-specific knockout.** (A) ZFNs specifically targeting the AT1R$_A$ (*Agtr1a*) gene were designed in a region of exon 3 where specific sequence differences (underlined) would prevent ZFN binding and mutagenesis of AT1R$_B$ (*Agtr1b*). The ZFN target is in bold and the specific monomer binding sites are underlined. A frame-shifting 2-bp deletion results in the formation of an *Acc*I restriction endonuclease site specifically in the AT1R$_A$ mutant (boxed). (B) The frameshift results in an predicted early truncation of the nascent AT1R$_A$ peptide containing 87 normal amino acids and harboring only the first two of seven transmembrane domains, followed by 13 nonsense amino acids. (C) PCR on cDNA reveals the expected amplification of AT1R$_A$ and AT1R$_B$ mRNAs with an expected product of 171-bp (expected 169-bp for the AT1R$_A$ mutant). *Acc*I restriction endonuclease specific cleavage of the 169-bp mutant AT1R$_A$ transcript into two fragments of 125 and 44-bp in AT1R$_A$ mutant rats (SS-AT1KO), but not control SS animals (SS-AT1WT).

## Physiological verification of selective mutation of the AT1R$_A$ receptor

**Fig 2A and 2B** shows acute blood pressure response to a bolus dose of AngII. Rats anesthetized with pentobarbital were catheterized and baseline blood pressure was established. A 100uL bolus of saline containing 0.32μg/kg AngII was administered via tail vein injection. A 100uL bolus of saline alone produced no response. SS-AT1WT rats demonstrated a robust increase in blood pressure in response to AngII (average change of 46.4 mmHg; p = 0.000392 via paired t-test), and this response was absent in SS-AT1KO rats (average change of 0.82 mmHg; p = 0.56 via paired t-test).

## AT1R$_A$ mutation does not alter expression of the Mas receptor or AT1R$_B$

To determine whether altered receptor expression could account for the observed phenotypes, we examined receptor expression in SS-AT1WT and SS-AT1KO cells. No difference in Mas1 expression was detected between SS-AT1WT and SS-AT1KO cells, suggesting that altered levels of Mas1 do not account for phenotypic changes observed in this study (**Fig 2C**; p = 0.211 via 2-tailed t-test). Western blot confirmed the presence of Mas1 protein in both SS-AT1KO and SS-AT1WT endothelial cells (Fig 2D; complete blot found in S1 raw images). No differences were detected in Mas1 protein expression between the two groups (Fig 2E; p = 0.44 via 2-tailed t-test).

Because rats express two forms of AT1R, we examined whether expression of AT1R$_B$ was detectable in either SS-AT1WT or SS-AT1KO endothelial cells. Consistent with previous studies suggesting little to no expression of AT1R$_B$ in microvasculature[19], AT1R$_B$ was undetectable in both cell types. Note that expression of AT1R$_A$ is detectable but decreased (p = 0.00283

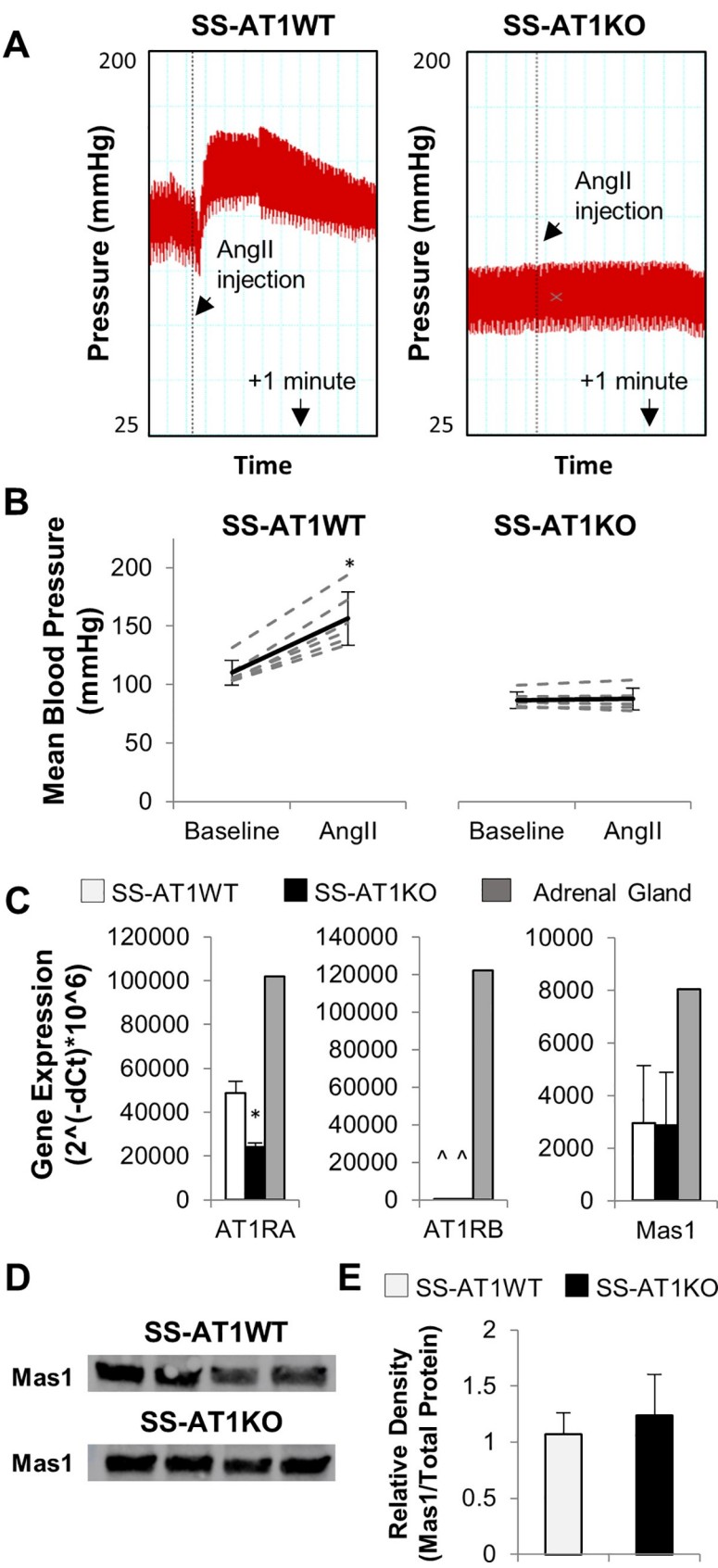

Fig 2. Physiological validation of AT1R$_A$ knockout model. Angiotensin II bolus (0.32ug/kg i.v.) causes an acute increase in blood pressure in SS-AT1WT but not SS-AT1KO rats. A. Representative acute blood pressure tracings. B. Baseline and peak blood pressure from all rats measured. Dashed lines represent individual animals while solid line represents mean +/- standard deviation (* p < 0.05 vs baseline via paired t-test). C. Expression of AT1RA, AT1RB and Mas1 was assessed in SS-AT1WT and SS-AT1KO endothelial cells. AT1RA expression was decreased in SS-AT1KO cells, consistent with increased turnover of degenerate transcripts (* p < 0.05 vs SS-AT1WT via Students t-test; ^ not detected). AT1RB was not detected in either group, and Mas1 expression was not altered by mutation of AT1RA. Hypoxanthine-guanine phosphoribosyltransferatse 1 (HPRT1) was used as the control for dCT calculation. SS adrenal gland, a tissue known to express all three receptors, was used as a positive control. D. Western blot for Mas1. Mas1 was present in both SS-AT1WT and SS-AT1KO endothelial cells; each lane contains protein from a separately cultured plate of endothelial cells. E. Relative quantification of Mas1 in SS-AT1WT and SS-AT1KO EC lysates. No difference was detected between these groups (n = 4; p = 0.442 via Students t-test).

via 2-tailed t-test), consistent with production and degradation of degenerate AT1R$_A$ transcript.

## AT1R$_A$ mutation abolishes the proangiogenic effects of Ang-(1–7) *in vivo*

We have previously shown that 7 days of muscle contraction induced by electrical stimulation of the hind limb *in vivo* (ESTIM) produces a robust angiogenic response that is depressed in the SS rat [1,17] and that chronic low-dose Ang-(1–7) infusion restores the angiogenic response in these animals[1]. To determine whether AT1R$_A$ was necessary for this effect, we performed ESTIM in SS-AT1WT and SS-AT1KO rats receiving low-dose Ang-(1–7) infusion (**Fig 3**). ESTIM in combination with vehicle treatment (without low-dose Ang-(1–7)) did not produce a significant increase in hind limb vessel density in either group (WT p = 0.1319; KO p = 0.3803). Ang-(1–7) enhanced the effects of electrical stimulation in the SS-AT1WT group (p = 0.0053) but had no effect in SS-AT1KO animals (p = 0.2582), suggesting that the AT1R$_A$ is necessary for the effects of Ang-(1–7) on angiogenesis in response to electrical stimulation. Note that in our previous work, competitive antagonism at AT1R$_A$ with losartan did not affect the ability of Ang-(1–7) to enhance angiogenesis[3]. This suggests that the observed phenotype is independent of ligand binding to the AT1R.

## AT1R$_A$ mutation impairs the ability of Ang-(1–7) to enhance endothelial cell tube formation *in vitro*

We have previously shown that Ang-(1–7) enhances the angiogenic capability of endothelial cells *in vitro* [3]. To determine whether AT1R$_A$ contributes to this effect, tube formation in SS-AT1WT cells and in SS-AT1KO cells was evaluated for 48 hours in the presence or absence of Ang-(1–7), the Mas1 antagonist A779, the AT1R antagonist Losartan, and VEGF, a positive control (**Fig 4**). Ang-(1–7) stimulated a significant increase in tube formation at both 24 hours and 48 hours in SS-AT1WT cells but not in SS-AT1KO cells. This effect was attenuated by antagonism of Mas1 but not affected by antagonism of AT1R. Both cell types demonstrated increased tube formation in response to VEGF, demonstrating that SS-AT1KO endothelial cells are able to respond to proangiogenic stimuli other than Ang-(1–7).

## AT1R$_A$ mutation alters the transcriptional response to Ang-(1–7) in endothelial cells

We previously observed that Ang-(1–7) treatment resulted in increased expression of proangiogenic transcripts in SS endothelial cells[3]. We compared the expression profile of angiogenic transcripts in Ang-(1–7) and vehicle treated SS-AT1WT and SS-AT1KO endothelial cells via qPCR. The trend in the overall dataset is for an Ang-(1–7) induced increase in expression in angiogenesis related transcripts in SS-AT1WT cells but not SS-AT1KO cells (**Fig 5**; R$^2$

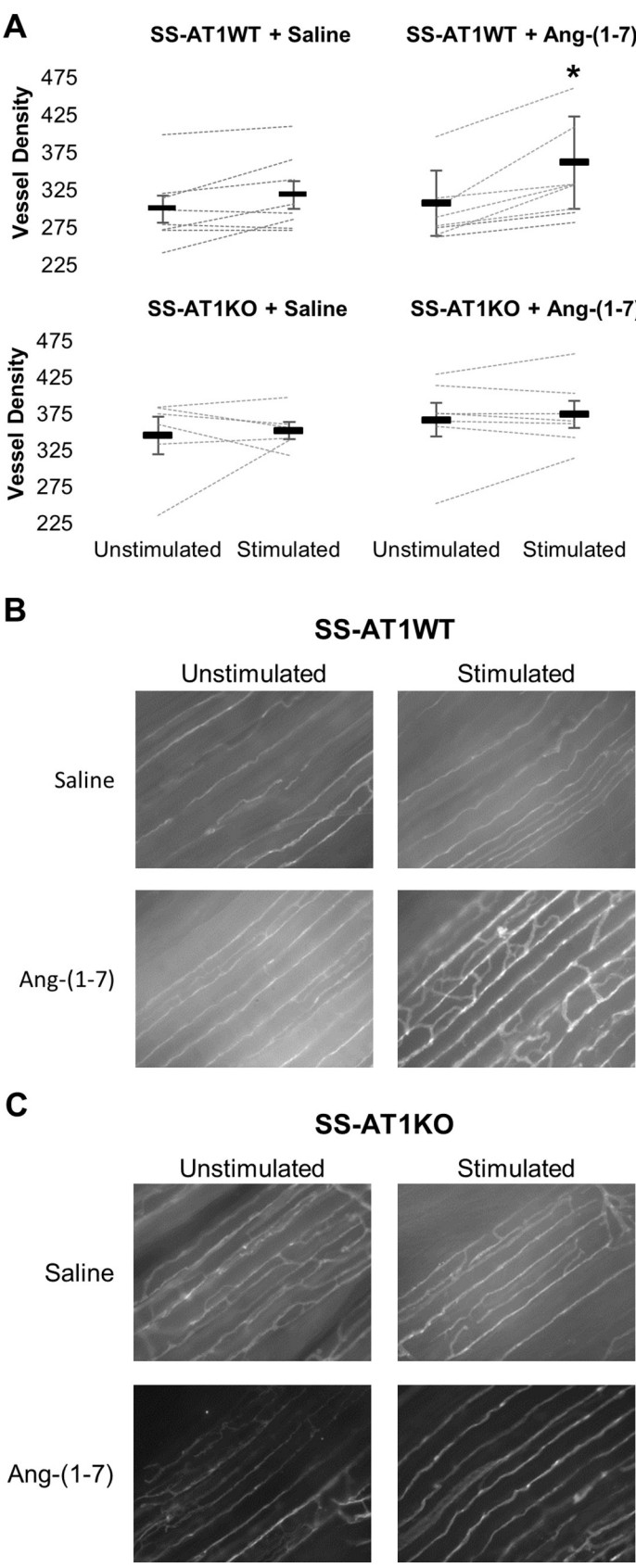

**Fig 3. Angiotensin-(1–7) enhances angiogenesis in SS-AT1WT but not SS-AT1KO rats.** Rats underwent electrical stimulation of one hind-limb for 7 days while infused with saline or 3ng/kg/min Ang-(1–7) via jugular catheter. A. Plots of vessel density of stimulated and unstimulated hind limbs. Dashed lines represent individual rats while bolded bars represent mean +/- standard deviation (* p < 0.05 vs unstimulated limb via paired t-test). B-C. Representative vessel density images from each group. ^ Units are vessel-grid intersections per microscope field as described in the methods.

= 9.2x10$^{-5}$). Hypothesis testing via Deming regression showed that the slope representing the relationship between SS-AT1WT and SS-AT1KO values was different from 1 (p < 0.0005) but not different from 0 (p = 0.87).

Of 84 genes that were tested, 19 were excluded from our analysis due to amplification failure in 2 or more biological replicates within a single experimental group. These data are available in **S5 Table**. Ang-(1–7) treatment resulted in significant changes in the expression of 9 genes in SS-AT1WT cells but only 1 transcript in SS-AT1KO cells (**Table 1**). After correction for multiple comparisons via modified Hochberg step-up procedure[31] with FDR set at 0.05,

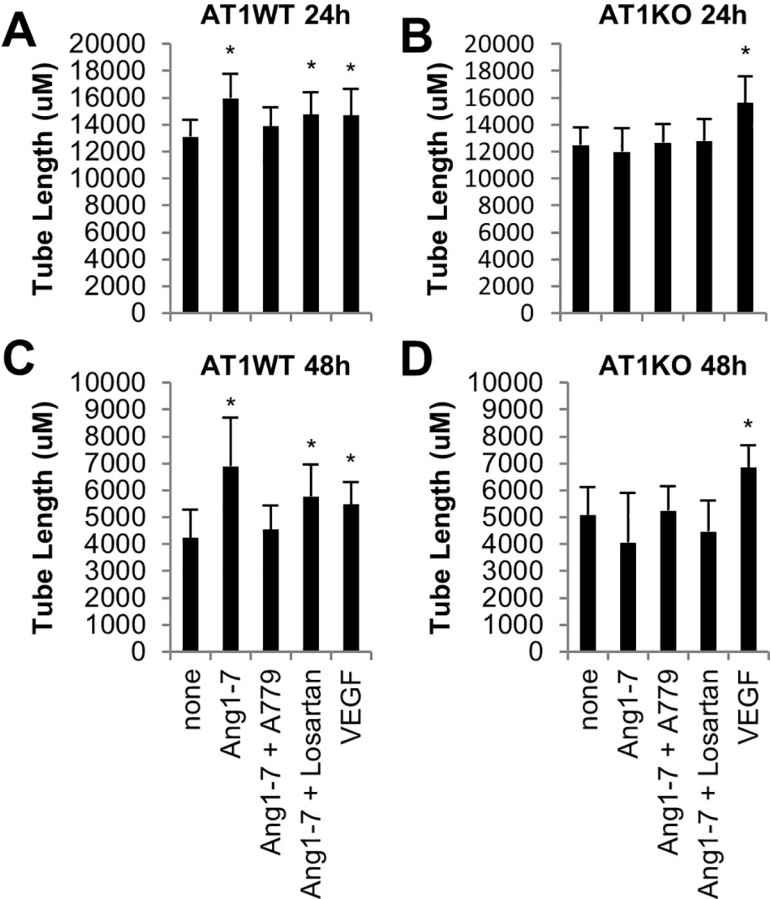

**Fig 4. Angiotensin-(1–7) enhances endothelial cell tube formation in SS-AT1WT but not SS-AT1KO endothelial cells.** Endothelial cells spontaneously form tube-like structures when cultured in a basement-membrane like matrix. A. SS-AT1WT endothelial cell tube length at 24 hours. B. SS-AT1KO endothelial cell tube length at 24 hours. C. SS-AT1WT endothelial cell tube length at 48 hours. D. SS-AT1KO endothelial cell tube length at 48 hours. Bars represent mean +/- standard deviation (* p < 0.05 vs none via 1-way ANOVA with Dunnett's Method for multiple comparisons). None = control, Ang1-7 = angiotensin-(1–7), A779 = Mas1 antagonist, Losartan = AT1 antagonist, VEGF = vascular endothelial growth factor.

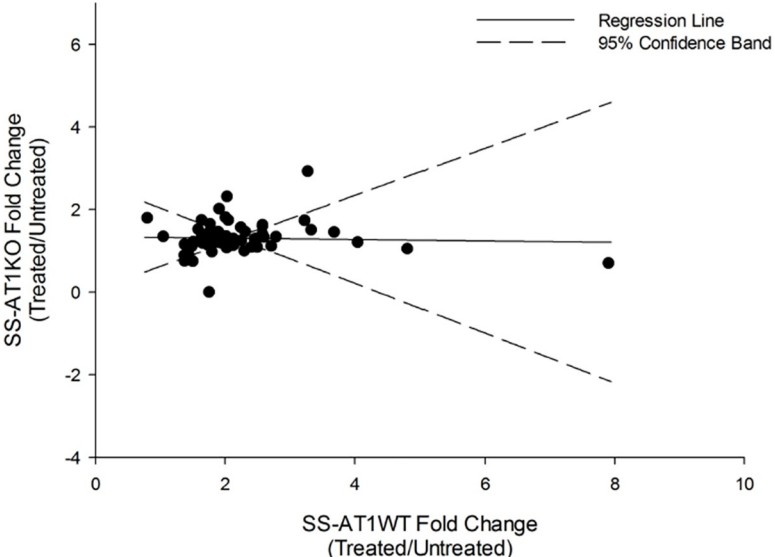

**Fig 5. AT1R$_A$ contributes to the transcriptional response to Ang-(1–7).** Levels of mRNA associated with angiogenesis pathways increased upon stimulation with Ang-(1–7) in Dahl salt-sensitive endothelial cells but not AT1a knockout cells. Values represent fold change in expression as detected via qPCR, with SS-AT1WT value on the x-axis and SS-AT1KO values on the y-axis ($R^2 = 9.2 \times 10^{-5}$). Hypothesis testing via Deming regression showed that the slope representing the relationship between SS-AT1WT and SS-AT1KO values was different from 1 ($p < 0.0005$) but not different from 0 ($p = 0.87$).

significant changes were detected in the expression of 5 genes in SS-AT1WT cells but only 1 transcript in SS-AT1KO cells.

**Table 1. Analysis of an angiogenesis RT-PCR gene expression array following Ang-(1–7) stimulation of rat microvascular endothelial cells.**

| Gene | Protein Annotation | Fold Regulation (SS)* | p-value (SS) | Fold Regulation (AT1KO)* | p-value (AT1KO) | Notable Signaling Involvement |
|---|---|---|---|---|---|---|
| Akt1 | AKT serine/threonine kinase 1 | 1.54 | 0.030 | 1.21 | 0.396 | PDGF, cell survival, angiogenesis, insulin signaling |
| Fgfr3 | Fibroblast growth factor receptor 3 | 1.89 | 0.006^ | 1.36 | 0.016^ | PI3K/AKT activation, angiogenesis, apoptosis |
| Il1b | Interleukin-1 beta | 2.03 | 0.034 | 2.32 | 0.080 | NF-kappaB signaling, MAPK/ERK, JAK-STAT, AKT, TLR signaling |
| Mmp2 | 72 kDa type IV collagenase | 2.00 | 0.029 | 1.22 | 0.126 | adhesion, angiogenesis, Tie2 signaling, immune cell transmigration |
| Ptgs1 | Prostaglandin G/H synthase 1 | 1.66 | 0.019^ | 1.19 | 0.132 | prostaglandin signaling |
| Ptk2 | Focal adhesion kinase 1 | 1.73 | 0.032 | 1.15 | 0.063 | migration, PI3K, AKT, MAPK/ERK, Rho GTPase signaling |
| Tgfb3 | Transforming growth factor beta-3 | 2.13 | 0.003^ | 1.15 | 0.331 | p38-MAPK, angiogenesis, Wnt/Hedgehog/Notch |
| Tgfbr1 | TGF-beta receptor type-1 | 2.09 | 0.000^ | 1.29 | 0.093 | angiogenesis, Wnt/Hedgehog/Notch, apoptosis, AKT, NF-kappaB |
| Thbs1 | Thrombospondin-1 | 2.12 | 0.003^ | 1.13 | 0.065 | TGF-beta, angiogenesis, adhesion, PI3K/AKT |

Table includes all genes with significant fold change in gene expression in at least one strain ($p \leq 0.05$; paired t-test within plate); biologic processes include but are not limited to those above (N = 3).

*Indicates 100 nM Ang-(1–7) stimulated versus unstimulated endothelial cells.

^ Indicates that p-value remained significant after correction for multiple comparisons via modified Hochberg step-up procedure[31] with FDR set at 0.05.

## AT1R$_A$ mutation alters the protein complex formed upon binding of Ang-(1–7) to the Mas receptor

Previously, we identified components of the Ang-(1–7)/Mas1 signaling complex[3]. Using this data as a baseline for analysis, we performed a new experiment using tandem mass spectrometry-based proteomics to identify molecules upregulated in, or unique to, the Ang-(1–7) activated Mas1 signaling complex in SS-AT1WT endothelial cells but not in SS-AT1KO cells. Because AT1R is a component of the Ang-(1–7)/Mas1 signaling complex in SS-AT1WT cells [3] and physiological and transcriptional responses to Ang-(1–7) are altered in SS-AT1KO cells (**Figs 3–5**), we hypothesized that the Ang-(1–7)/Mas1 signaling complex would be altered in SS-AT1KO cells.

The technique used here is cryolysis followed by co-immunoprecipitation as described in our previous study [3] and in the reagent manufacturer's user guide (Thermo Fisher 14321D). Cryolysis stabilizes large protein complexes, facilitating their isolation together and promoting minimal loss in the complex. This provides a deeper magnitude of identification into the protein binding complex compared to simple co-immunoprecipitation. Note that this means identified proteins do not all bind directly to the target, in this case Mas1, but rather that they are part of a complex of interacting proteins.

**Table 2** shows the complete list of proteins that met our criteria as AT1R related components of the Ang-(1–7)/Mas1 signaling complex: FDR < 5%, statistically significant difference between treated and untreated experimental groups (normalized p-value < .05), at least a 2-fold enrichment in the Ang-(1–7) treated group, presence in greater than 50% of replicates in at least one experimental group, at least 6 scans detected, and upregulation in the SS-AT1WT comparison but not the SS-AT1KO comparison. Note that several of these proteins are part of the previously described Ang-(1–7)/Mas1 signaling complex[3]. The proteomic datasets can be found in supplemental materials (**S3 Table** and **S4 Table**).

**Fig 6** shows a simplified version of the Ang-(1–7)/Mas signaling pathway previously described [3]. Proteins and mRNA transcripts previously identified in the Mas signaling pathway and associated with AT1R$_A$ based on the present study are denoted in color while other previously identified components are represented in grey. Of note are the NOTCH family of proteins and protein kinase D1, both of which were identified in the Ang-(1–7)/Mas1 signaling complex previously and are related to known angiogenesis signaling pathways involving ERK, p38-MAPK, and/or Akt[3].

This data may also be useful in developing hypotheses regarding potential AT1R independent actions of Ang-(1–7) via Mas1. **S2 Table** shows proteins that may be related to AT1R independent signaling downstream of the Ang-(1–7) complex. This table was formed by applying the following criteria: FDR < 5%, statistically significant difference between treated and untreated experimental groups (normalized p-value < .05), at least a 2-fold enrichment in the Ang-(1–7) treated group, presence in greater than 50% of replicates in at least one experimental group, at least scans detected, and upregulation in the SS-AT1KO comparison but not the SS-AT1WT comparison. Of particular interest may be PAK4 (log2ratio = 4.09; p = 3.31x10$^{-5}$), which promotes cell survival [32,33], affects cell adhesion [34] and is implicated in the epithelial-mesenchymal transition in cancer [35], and PA2G4 (log2ratio = 2.58; p = 0.046), a DNA binding protein [36] which is implicated in both development [37] and a variety of cancers [38,39].

## Discussion

The current study provides both physiological and biochemical support for an interaction between AT1R and Mas1 signal transduction. It has been previously hypothesized that AT1R

**Table 2. Ang-(1–7) stimulated MAS1 receptor immuno-precipitation divergent 'top proteomic hits'.**

| Accession Number | Annotated Protein | NormLog2Ratio* | norm. p-value* | Notable signaling involvement |
|---|---|---|---|---|
| Q8BIZ0 | Protocadherin-20 (Pcdh20) | Unique | 3.63E-08 | Calcium-dependent cell-adhesion |
| Q63532 | Cornifin-A (SPR1A) | Unique | 1.42E-07 | Membrane cross-linking |
| Q62101 | Serine/threonine-protein kinase D1 (PRKD1) | Unique | 6.93E-05 | PKC (+), DAG (+), ERK1/2 (+), IKK/NFkB (+), p38MAPK (+), AKT (+) and EGF (-) Signaling |
| Q80UN1 | BTB/POZ domain-containing protein KCTD9 | Unique | 2.38E-03 | Protein ubiquitination |
| Q9QYR6 | Microtubule-associated protein 1A (Map1a) | Unique | 4.90E-03 | Structural protein |
| Q99466^ | Neurogenic locus notch homolog protein (NOTCH) family^ | Unique | 2.16E-02 | Cell Survival Signaling (+), angiogenesis |
| P04095 | Proliferin-1 precursor (Mitogen-regulated protein 1) | 4.36 | 2.39E-06 | Growth factor and/or angiogenesis factor |
| O35625 | Axis inhibition protein 1 (Axin-1) | 4.15 | 1.63E-05 | Wnt-signaling modification; JNK signaling |
| Q7TPH6 | E3 ubiquitin-protein ligase MYCBP2 | 3.60 | 1.80E-06 | Ubiquitination; transcriptional regulator of MYC |
| Q9JK88 | Serpin I2 | 3.60 | 7.35E-04 | Endopeptidase inhibitor |
| Q01887 | Tyrosine-protein kinase RYK | 3.27 | 5.83E-07 | Wnt coreceptor |
| Q9JHZ9 | Sodium-coupled neutral amino acid transporter 3 (Slc38a3) | 1.93 | 8.21E-06 | Sodium-dependent aa/proton transporter |
| Q6P542 | ATP-binding cassette sub-family F member 1 (Abcf1) | 1.95 | 5.80E-13 | mRNA translation initiation; not ribosome biogenesis |
| P48744 | Norrin precursor | 1.60 | 1.61E-05 | Wnt signaling; retinal vascularization |

All proteins indicated passed all stringent filters indicated in the Methods; full protein lists can be found in S3 Table and S4 Table.

*Mas1 IP: 100 nM Ang-(1–7) stimulated SS EC versus unstimulated SS EC (Per condition: N = 3; 6 total runs)

^Peptides mapped to multiple members of the NOTCH family; accession for NOTCH4 was used based on previous findings [3].

and Mas1 heterodimerize[12]. However, the mechanism and functional effects of this potential interaction are still being examined. Kostenis and colleagues showed that AT1R and Mas co-localize using bioluminescence resonance energy transfer (BRET) and observed that expression of Mas decreased AngII signaling despite an increase in AngII binding capacity[12]. It has been shown that Mas receptor knockout has little to no effect on the ability of AT1R to bind AngII, suggesting that competitive binding of AngII to Mas is unlikely to account for the decreased AngII signaling observed by Kostenis and colleagues[5]. Another study observed rescue of function of a mutant AT1R by Mas1 expression, which altered the distribution of the mutant AT1R within the cell[13]. These findings support the hypothesis that dimerization occurs between AT1R and Mas1.

While the mechanism and conformation of AT1R and Mas1 interaction at the molecular level must still be examined directly, the current results strongly suggest that interaction of AT1R$_A$ and Mas1 is necessary for the transduction of signals downstream of Ang1-7/Mas1 binding in endothelial cells. Previously, AT1R$_A$ was identified in the immunoprecipitated Ang1-7/Mas1 protein complex [3], suggesting physical proximity. This identification only occurred when Mas1 was bound to its ligand, Ang1-7, suggesting a functional interaction. Several downstream proteins identified in the Ang1-7/Mas1 protein complex are known mediators of AT1R$_A$ signaling, consistent with a transactivation-like signaling event that requires interaction between AT1R$_A$ and Mas1 to achieve complete signal transduction. In the present study, absence of AT1R$_A$ abolishes the physiological effect of Ang1-7 both *in vivo* and *in vitro* (**Figs 3 and 4**) as well as the transcriptional response to Ang-(1–7), supporting the hypothesis that proangiogenic Ang1-7/Mas1 signaling is altered in the absence of AT1R$_A$ (**Table 1; Fig 5**).

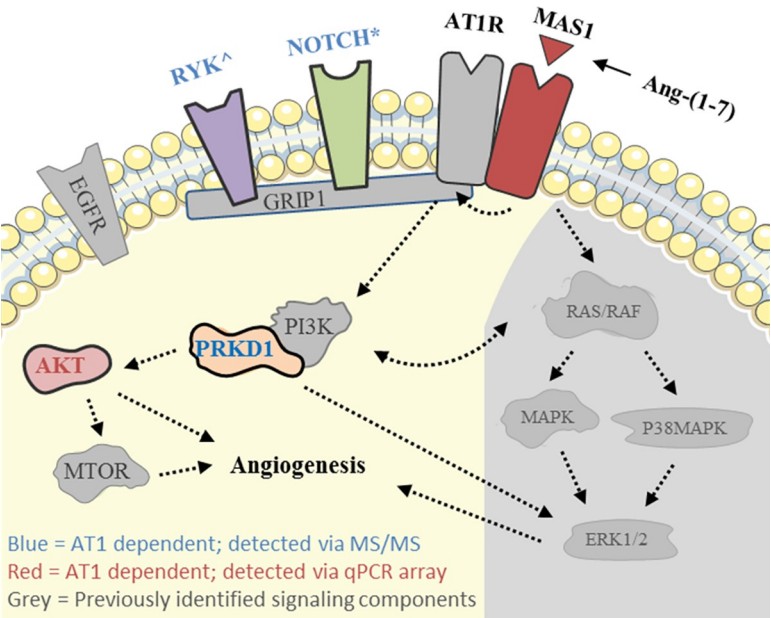

**Fig 6. Diagram of suggested AT1$_A$ dependent components of the Ang1-7/Mas signaling complex.** Unless otherwise noted, molecules represented here were previously identified as components of the Ang-(1–7)/Mas1 signaling complex. Colored molecules represent AT1R$_A$ dependent components detected in the current study via mass spectrometry-based proteomics (blue label) and/or qPCR (red label). *Peptides mapped to NOTCH were not unique to specific NOTCH family members. ^RYK was not detected in our previous study but was included here due to known links to angiogenesis.

In further support of this hypothesis, the composition of the Ang1-7/Mas1 protein complex is altered in the absence of AT1R$_A$ (**Table 2**).

Changes in the Ang-(1–7)/Mas signaling complex in the absence of intact AT1R suggest avenues for further study. Several candidates are identified in **Fig 6**. These potential AT1R-dependent members of the Mas1 signaling complex have previously established roles in angiogenesis, as outlined below.

Protein kinase D1 (PRKD1) is implicated here and is known to affect angiogenesis in several ways. In human umbilical vein endothelial cells, PRKD1 is required for VEGF-induced angiogenesis mediated by histone deacetylase 5 (HDAC5) [40]. Furthermore, knockdown of PRKD1 inhibits physiological angiogenesis and abolishes tumor angiogenesis in zebrafish [41]. Akt, a known regulator of angiogenesis, is in turn regulated by PRKD1 downstream of G protein-coupled receptors [42,43]. AngII is known to activate PRKD1[44], which has been shown to induce phosphorylation of ERK and increase the duration of ERK activation[45]. In the context of our studies, it is possible that Ang-(1–7)/Mas1 regulates Akt, ERK and/or HDAC5 via AT1R activation of PRKD1.

Our data implicate the Notch family of signaling components as a proangiogenic signaling mechanism in endothelial cells downstream of the Mas1/AT1R$_A$ complex. Notch proteins are known to play a complex regulatory role in the formation of new blood vessels. Functions of Notch signaling include cell type specification, proliferation, and vessel stability, among others [46]. For example, activated Notch4 inhibits human microvascular endothelial cell sprouting in response to FGF-2 and VEGF [47], and constitutively active Notch4 has been shown to inhibit endothelial cell apoptosis [48]. Previous studies in our laboratory identified Notch4 as a member of the Ang-(1–7)/Mas signaling complex in endothelial cells [3] and Notch3 as a potential mediator of endothelial progenitor cell (EPC) dysfunction [49]. EPCs are bone-

marrow derived stem-cells that contribute to therapeutic angiogenesis in animal models [50]. The therapeutic efficacy of EPCs in the SS rat is compromised, and our previous study suggests that this dysfunction is linked to altered methylation of Notch3 that leads to suppression of Notch3 expression [49].

RYK, a receptor for Wnt5a, was also suggested to be an AT1R dependent member of the Ang-(1–7)/Mas signaling complex. Wnt5a signaling is known to enhance the proliferation, survival, and migration of endothelial cells[51,52]. In addition, expression of transforming growth factor beta-3 (Tgfb3) and TGF-beta receptor type-1 (Tgfbr1) transcripts—both of which are involved in crosstalk with Wnt signaling[53,54]—was upregulated after Ang-(1–7) treatment in SS-AT1WT but not SS-AT1KO endothelial cells. Thus, despite lack of a more direct link between AT1R, Mas1, and RYK, Wnt signaling via RYK represents another strong candidate for further study.

The present study also identified several other well-known mediators of proangiogenic signaling that are responsive to Ang-(1–7) as potentially AT1R$_A$ dependent, by either transcriptional changes or implied pathway identification. It is important to note that altered mRNA levels are likely to be due to altered transcription, trafficking, or degradation, all of which are downstream effects of signaling *per se*. Thus, molecules downstream of signaling components identified via mass spectrometry, such as Akt, could contribute to signaling while those identified only by qPCR, such as Fgfr3 and Mmp2, may represent effects of signaling rather than mediators of signal transduction.

An important limitation of this study is the assumption that Ang-(1–7) does not exert its proangiogenic effects via binding to AT1R. This assumption is based upon our previous finding that the proangiogenic effects of Ang-(1–7) are blocked by the Mas1 antagonist A779 but not by the AT1R antagonist losartan [3]. It is possible that yet to be understood complexity will invalidate this assumption. For instance, an altered conformation of heterodimerized AT1R could allow Ang-(1–7) to bind AT1R despite the presence of losartan. However, note that the convergent, proangiogenic effects of AngII in our model were attenuated in the presence of losartan [3]. Unless AngII and Ang-(1–7) act via separate sets of AT1R, binding of AT1R by Ang-(1–7) in the presence of losartan is an unlikely mechanism for these losartan-independent proangiogenic effects. Given that the strongest evidence for Ang-(1–7) binding AT1R is displacement of AngII [55,56], we do not believe that binding of Ang-(1–7) to a separate set of AT1R that are unaffected by losartan is responsible for the proangiogenic effects observed in our model. Note that we do not discount AT1R binding as a mechanism for other effects mediated by Ang-(1–7). In fact, this possibility is of ongoing interest in the field. For instance, there is evidence that Ang-(1–7) can act as a biased ligand in cells expressing AT1R but not Mas1 [57]. There is also recent work suggesting that Ang-(1–7) does not interact directly with Mas1 [58]. This is in conflict with work demonstrating that Ang-(1–7) exerts effects via Mas1 [5], including in cells expressing Mas1 but not AT1R [59] and work that demonstrates Ang-(1–7) induced Mas1 internalization [60]. Given the ever-growing complexity of the renin-angiotensin system [61,62], including transactivation of Mas1 downstream of AngII [63], it seems unlikely that the effects of Ang-(1–7) are mediated via a single receptor. It will be important to reassess this work and other work in the field as our understanding of this complexity grows.

These data may be useful in developing hypotheses regarding AT1R independent Mas1 signaling. Given that the remainder of the data presented here examines only AT1R dependent effects of Ang-(1–7)/Mas1 signaling, the conclusions that can be drawn are limited. With this limitation in mind, several proteins of interest were identified as potential AT1R independent mediators of Ang-(1–7)/Mas1 signaling (**S2 Table**). Mas1 was first described as an oncogene [64] and continues to be of interest in cancer [65,66]. Therefore, PAX4 and PA2G4, both of

which are implicated in the development of cancer [35,38,39] and were identified here as a potential mediators of AT1R independent Ang-(1–7)/Mas1 signaling, represent ideal candidates for further study. In particular, PAK4, which regulates cell adhesion [34] and apoptosis [32], is of interest in this context given the known effects on cell growth and/or metastasis by Ang-(1–7)/Mas1 in nasopharyngeal carcinoma [67], prostate cancer [68,69], breast cancer [70], and other cancers [71].

This study, combined with our previous work showing the presence of AT1R in the activated Ang-(1–7)/Mas signaling complex, suggests that physical interaction of Mas1 and AT1R is important for signal transduction. Future work will seek to identify AT1R/Mas1 conformations consistent with attributes consistent with heterodimerization or oligomerization. Understanding of this protein complex at a structural level will allow the creation of higher resolution models to examine the effect of disrupted dimerization in the presence of otherwise functional receptors and enable further investigation of genetic differences that could lead to increased cardiovascular risk via disruption of AT1R/Mas1 binding.

## Supporting information

**S1 Raw Images. Complete blot for Fig 2D.**
(PDF)

**S1 Table. CT values for Fig 2C.**
(XLS)

**S2 Table. Proteins upregulated in the SS-AT1KO Ang-(1–7)/Mas1 complex.**
(XLS)

**S3 Table. Proteomic Data for SS-AT1WT ECs +/- Ang-(1–7).**
(XLS)

**S4 Table. Proteomic Data for SS-AT1KO ECs +/- Ang-(1–7).**
(XLS)

**S5 Table. CT values for Table 1.**
(XLSX)

## Acknowledgments

The authors thank Katie Fink and Jessica R.C. Priestly for their valuable assistance in this project.

## Author Contributions

**Conceptualization:** Eric C. Exner, Timothy Stodola, Andrew S. Greene.

**Data curation:** Eric C. Exner.

**Formal analysis:** Eric C. Exner, Aron M. Geurts, Marc Casati.

**Funding acquisition:** Eric C. Exner, Aron M. Geurts, Brian R. Hoffmann, Julian H. Lombard, Andrew S. Greene.

**Investigation:** Eric C. Exner, Brian R. Hoffmann, Marc Casati, Julian H. Lombard.

**Methodology:** Eric C. Exner, Aron M. Geurts, Brian R. Hoffmann, Marc Casati, Julian H. Lombard.

**Resources:** Aron M. Geurts, Brian R. Hoffmann, Julian H. Lombard, Andrew S. Greene.

**Software:** Eric C. Exner.

**Supervision:** Julian H. Lombard, Andrew S. Greene.

**Validation:** Eric C. Exner, Nikita R. Dsouza, Michael Zimmermann.

**Visualization:** Eric C. Exner.

**Writing – original draft:** Eric C. Exner.

**Writing – review & editing:** Eric C. Exner, Aron M. Geurts, Brian R. Hoffmann, Marc Casati, Timothy Stodola, Nikita R. Dsouza, Michael Zimmermann, Julian H. Lombard, Andrew S. Greene.

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
