## [Decision Letter · Decision Letter 0]

2 Dec 2019

PONE-D-19-31333

Interaction between Mas1 and AT1RA contributes to enhancement of skeletal muscle angiogenesis by angiotensin-(1-7) in Dahl salt-sensitive rats

PLOS ONE

Dear Dr. Exner,

Thank you for submitting your manuscript to PLOS ONE. After careful consideration, we feel that it has merit but does not fully meet PLOS ONE’s publication criteria as it currently stands. Therefore, we invite you to submit a revised version of the manuscript that addresses the points raised during the review process.

We would appreciate receiving your revised manuscript by Jan 16 2020 11:59PM. To enhance the reproducibility of your results, we recommend that if applicable you deposit your laboratory protocols in protocols.io, where a protocol can be assigned its own identifier (DOI) such that it can be cited independently in the future. For instructions see: http://journals.plos.org/plosone/s/submission-guidelines#loc-laboratory-protocols

We look forward to receiving your revised manuscript.

Kind regards,

Michael Bader

Academic Editor

PLOS ONE

Journal Requirements:

Reviewers' comments:

Reviewer's Responses to Questions

**Comments to the Author**

1. Is the manuscript technically sound, and do the data support the conclusions?

Reviewer #1: Partly

Reviewer #2: Yes

2. Has the statistical analysis been performed appropriately and rigorously? 

Reviewer #1: I Don't Know

Reviewer #2: Yes

3. Have the authors made all data underlying the findings in their manuscript fully available?

Reviewer #1: Yes

Reviewer #2: Yes

4. Is the manuscript presented in an intelligible fashion and written in standard English?

Reviewer #1: Yes

Reviewer #2: Yes

5. Review Comments to the Author

Reviewer #1: Exner et al. study the role of the interaction between Mas1 and AT1 receptor in the angiogenic effect of Ang-(1-7). This is done in a rat model of hindlimb angiogenesis. The rats are of the Dahl SS strain, and harbour a mutant AT1A receptor that is dysfunctional. Normally, SS rats have no significant angiogenesis upon electrical stimulation, but parallel infusion of Ang-(1-7) results in neovascularization. The mutant AT1A receptor abolishes this effect of Ang-(1-7). Similar observations were made in cultured EC. Signaling is changed, and the Mas1 – AT1 R complex is altered by the mutation.

The paper confirms that Mas ATR1 R interaction rather than Mas receptor alone determines responses to Ang-(1-7). Thus, the paper supports the work that is being performed to unravel the physiological relevance of Ang-(1-7) and its clinical translation.

Major concerns:

Fig. 2C. There appears to be no mRNA of Mas1 present in EC’s. What were the number of PCR cycles that was needed to detect a signal? And was there any Mas1 protein? This needs to be answered. It should be possible to show protein levels since the authors used antibodies for immuno-precipitation of Mas1. Please also provide adrenal gland gene expression from both WT and KO.

It is a bit strange to observe that WT show no angiogenic response to electrical stimulation. It makes one wonder what we are looking at. What happens in WT rats (no SS)? Does electrical stimulation elicit angiogenesis in such a model, and is this promoted by Ang-(1-7) and AngII? Or do we per se need a low renin model? If so, why is that? Are sufficient amounts of free AT1R needed perhaps, if one wishes to observe any Ang-(1-7) effect (through intercation of Mas with AT1R, or even binding of Ang-(1-7) to AT1R)? In SS rats Ang II levels are indeed low. It is important to test this.

It is stated that the inability of losartan to affect Ang-(1-7) effects suggests that the ligand does not need to bind AT1R. This sounds logical when assuming that AT1R is in the same conformation as it would be when responsive to AngII. However, what if Mas1-associated AT1R has a different conformation, allowing activation by Ang-(1-7) (and antagonism by A779), and lowers losartan binding affinity? In fact, Mas was thought to be the AngII receptor when characterized as such for the first time in 1988 (Jackson et al. Nature 335). Probably Mas levels regulate AT1R levels and affinity? See also Gaidarov et al. Cell Signal. 2018, who present data that dismiss Mas as a binding partner for Ang-(1-7). This possibility should be taken into account. It can explain some of the older findings in which Ang-91-7) was identified as an AT1R antagonist (Mahon 1994, Gironacci 1999). In addition, the AT1R – Mas interaction appears to be important, and Ang-(1-7) is a tool to explore it. However, does this imply a physiological role for Ang-(1-7)? It is important to address these questions in the present manuscript. The comparison of Ang-(1-7) effect in Dahl SS vs. a WT rat strain might indicate if AT1R that is already occupied by AngII is in any way susceptible to modulation. If this is not the case it might be that Ang-(1-7) is indeed replacing AngII in SS rats to activate AT1R. In summary, it is still not clear if Mas is the direct binding partner of Ang-(1-7) (previous binding studies are not very convincing), but certainly the interaction of Mas with AT1R appears to be very important. The present and previous study of the authors indicate that AngII and Ang-(1-7) signaling pathways converge, providing evidence that the heptapeptide is in fact interacting with AT1R itself. Perhaps as a partial agonist, depending on the conformation of the receptor, which is regulated by Mas?

Table 1: As it appears no Bonferroni correction was used for the number of genes that was included. Should that not be done?

For studying the protein complex MAS1 was co-precipitated with proteins that were bound upon stimulation with Ang-(1-7). A lot of proteins appear to bind MAS1, which is puzzling. Is this in agreement with the structure of MAS1? Or do the various proteins complex which each other. In order to support these findings it seems necessary to explore in the available protein sequence libraries if certain sequences can explain the results. It might also help to distinguish the relevant findings from those that are merely supported by statistical analysis.

Minor concerns:

Legends are found in the middle of the body text. I guess they should be following at the end?

Reviewer #2: In this study, SS-rats with mutation of AT1 were created and the involvement of AT1 in A1-7/Mas induced angiogenesis was investigated. They found that pro-angiogenic response induced by angiotensin 1-7 in hind limb was absent in AT1-KO rats. Enhancement of EC tube formation by angiotensin 1-7 is similarly blunted in AT1-KO mutant ECs. Alternation of genes involved in angiogenesis by angiotensin 1-7 was mostly blunted in AT1-KO rats. They also found that AT1 mutation altered the protein complex formed by angiotensin 1-7 binding to Mas receptor. They concluded that these data support the hypothesis that interaction between AT1R and Mas1 contributes to proangiogenic signaling induced by angiotensin 1-7. These findings are interesting and would contribute to further understanding of the machinery of the receptor complex.

1. It is necessary to discuss about the proteins binding to Mas in response to A1-7 in AT1KO ECs. That could explain how the AT1-independent pathway of A1-7-Mas stimulation would work.

2. As related above, please specify if there any proteins binding to Mas in response to A1-7 in AT1KO ECs but not in WT ECs. If exists, please discuss about the functional relevance of the protein complex in the difference of signaling pathway between AT1KO and WT ECs.

3. It was reported that angiotensin 1-7 directly binds to AT1 (Hypertension. 2016 Dec;68(6):1365-1374.) It is necessary to discuss about the phenotype in AT1KO rats with respect to the direct effect of angiotensin 1-7 to AT1.

6. PLOS authors have the option to publish the peer review history of their article (what does this mean?). If published, this will include your full peer review and any attached files.

Reviewer #1: Yes: A.J.M. Roks

Reviewer #2: No

---

## [Author Response · Author response to Decision Letter 0]

26 Mar 2020

We appreciate the opportunity to resubmit this work - our responses can be found in the document entitled "Response to Reviewers".

---

## [Decision Letter · Decision Letter 1]

7 Apr 2020

Interaction between Mas1 and AT1RA contributes to enhancement of skeletal muscle angiogenesis by angiotensin-(1-7) in Dahl salt-sensitive rats

PONE-D-19-31333R1

Dear Dr. Exner,

We are pleased to inform you that your manuscript has been judged scientifically suitable for publication and will be formally accepted for publication once it complies with all outstanding technical requirements.

With kind regards,

Michael Bader

Academic Editor

PLOS ONE

Additional Editor Comments (optional):

Reviewers' comments:

Reviewer's Responses to Questions

**Comments to the Author**

1. If the authors have adequately addressed your comments raised in a previous round of review and you feel that this manuscript is now acceptable for publication, you may indicate that here to bypass the “Comments to the Author” section, enter your conflict of interest statement in the “Confidential to Editor” section, and submit your "Accept" recommendation.

Reviewer #1: All comments have been addressed

Reviewer #2: All comments have been addressed

2. Is the manuscript technically sound, and do the data support the conclusions?

Reviewer #1: Yes

Reviewer #2: Yes

3. Has the statistical analysis been performed appropriately and rigorously? 

Reviewer #1: Yes

Reviewer #2: Yes

4. Have the authors made all data underlying the findings in their manuscript fully available?

Reviewer #1: Yes

Reviewer #2: Yes

5. Is the manuscript presented in an intelligible fashion and written in standard English?

Reviewer #1: Yes

Reviewer #2: Yes

6. Review Comments to the Author

Reviewer #1: I have no further questions about this manuscript. All points were adequately addressed by the authors.

Reviewer #2: (No Response)

7. PLOS authors have the option to publish the peer review history of their article (what does this mean?). If published, this will include your full peer review and any attached files.

Reviewer #1: Yes: Anton J.M. Roks

Reviewer #2: No

---

## [Editor Report · Acceptance letter]

13 Apr 2020

PONE-D-19-31333R1 

Interaction between Mas1 and AT1RA contributes to enhancement of skeletal muscle angiogenesis by angiotensin-(1-7) in Dahl salt-sensitive rats 

Dear Dr. Exner:

I am pleased to inform you that your manuscript has been deemed suitable for publication in PLOS ONE. Congratulations! Your manuscript is now with our production department. 

With kind regards,

on behalf of

Prof. Michael Bader 

Academic Editor

PLOS ONE